# Ion solvation as a predictor of lanthanide adsorption structures and energetics in alumina nanopores

Anastasia G. Ilgen [1✉], Nadine Kabengi[2], Jacob G. Smith[1] & Kadie M. M. Sanchez[1]

Adsorption reactions at solid-water interfaces define elemental fate and transport and enable contaminant clean-up, water purification, and chemical separations. For nanoparticles and nanopores, nanoconfinement may lead to unexpected and hard-to-predict products and energetics of adsorption, compared to analogous unconfined surfaces. Here we use X-ray absorption fine structure spectroscopy and *operando* flow microcalorimetry to determine nanoconfinement effects on the energetics and local coordination environment of trivalent lanthanides adsorbed on $Al_2O_3$ surfaces. We show that the nanoconfinement effects on adsorption become more pronounced as the hydration free energy, $\Delta G_{hydr}$, of a lanthanide decreases. Neodymium ($Nd^{3+}$) has the least exothermic $\Delta G_{hydr}$ ($-3336$ kJ·mol$^{-1}$) and forms mostly outer-sphere complexes on unconfined $Al_2O_3$ surfaces but shifts to inner-sphere complexes within the 4 nm $Al_2O_3$ pores. Lutetium ($Lu^{3+}$) has the most exothermic $\Delta G_{hydr}$ ($-3589$ kJ·mol$^{-1}$) and forms inner-sphere adsorption complexes regardless of whether $Al_2O_3$ surfaces are nanoconfined. Importantly, the energetics of adsorption is exothermic in nanopores only, and becomes endothermic with increasing surface coverage. Changes to the energetics and products of adsorption in nanopores are ion-specific, even within chemically similar trivalent lanthanide series, and can be predicted by considering the hydration energies of adsorbing ions.

[1] Geochemistry Department, Sandia National Laboratories, 1515 Eubank Boulevard SE, Albuquerque, NM 87123, USA. [2] Department of Geosciences, Georgia State University, 24 Peachtree Center Avenue NE, Atlanta, GA 30303, USA. ✉email: agilgen@sandia.gov

Adsorption reactions at solid-water interfaces are relevant to ion-selective capture[1–7], chemical separations[8–11], catalysis[6], and environmental fate and transport[12–14]. When solid surfaces are nanoconfined (for nanoparticles or inside nanopores), their apparent reactivity starts being influenced by the surface energy terms, which become significant at the nano-scale[6,15–18]. Importantly, inside nanopores filled with aqueous solutions electrical double-layers (EDL) extending from the charged surfaces may overlap, causing the structures of nano-confined solutions to differ from those observed at unconfined surfaces. Nanoconfined water has a lower dielectric response[19–24], and lower density and surface tension[25], defined by the H-bonding structures[16,26] and slower rotational dynamics in near-interfacial regions[27,28]. Despite much information about nanoconfined water, the reactivities of nanoconfined surfaces, including the pathways and products of adsorption reactions, are not easily predicted[15,17,18,29–33]. This uncertainty is because adsorption in nanoconfined systems is dictated by both surface chemistry (e.g., the density of Si–OH, Al–OH, or other functional groups[17,18,30,31] and charge distribution[34]) as well as the size of the pore/channel that determines the solute structures within the overlapping EDLs. These interdependent effects have hindered the discovery of reliable predictors for how the energetics, path-ways, and products of adsorption reactions change under nanoconfinement[12,13,29,30]. Here we show that the hydration energy of an adsorbing ion ($\Delta G_{hydr}$) could be used as a mean-ingful predictor.

Previous studies have reported that nanoconfinement enhances inner-sphere adsorption and shifts the net adsorption heat from exo- (unconfined) to endothermic (nanoconfined) for cation adsorption onto negatively-charged silica (SiO₂) surfaces and on zeolites[15,29,30,32,35]. Ilgen et al. concluded that the adsorption of cations with less exothermic (less negative) $\Delta G_{hydr}$ is affected more by nanoconfinement compared to cations with more exo-thermic $\Delta G_{hydr}$ and tighter hydration shells[29]. This trend was observed for the products and heats of adsorption for SiO₂ sur-faces nanoconfined within pores under 7 nm in diameter[29]. Therefore, the $\Delta G_{hydr}$ of an ion could be used to anticipate the extent of nanoconfinement effects on its adsorption behavior. However, this trend has only been shown thus far for negatively-charged SiO₂ surfaces[29]. Here we study lanthanide adsorption on positively-charged alumina (Al₂O₃) surfaces and present further evidence that $\Delta G_{hydr}$ is in fact a reliable predictor of nanocon-finement effects on adsorption.

To test our hypothesis that *$\Delta G_{hydr}$ controls whether cation adsorption energetics and products are affected by nanoconfine-ment*, we exploit the gradual change in the $\Delta G_{hydr}$ of trivalent lanthanide cations (Ln³⁺) and compare their adsorption on unconfined Al₂O₃ (*i.e.*, corundum) and Al₂O₃ surfaces nano-confined within 4.4 nm pores. Using *operando* flow micro-calorimetry we show that at low surface coverages, adsorption reaction is endothermic for unconfined Al₂O₃ and becomes exothermic when Al₂O₃ surfaces are nanoconfined; adsorption becomes more endothermic as surface coverage is increased. Using X-ray absorption fine structure (XAFS) spectroscopy, we show that local structures around adsorbed neodymium (Nd³⁺) are vastly different for corundum and nanoconfined Al₂O₃ sur-face, while they are virtually indistinguishable on both surfaces for the stronger-hydrated lutetium (Lu³⁺). To our knowledge, this is the first study of lanthanide adsorption on nanoconfined Al₂O₃ surfaces that describes interfacial structures together with adsorption energetics.

Accurate molecular-scale descriptions of nanoconfined Al₂O₃ surface reactivities are crucial for predictive models of con-taminant mobilities, immobilization of radionuclides within heterogeneous nuclear wastes, and water purification with Al₂O₃

membranes[36]. Al₂O₃ is a building block of soils and rocks, therefore it can drive macroscopic chemical fluxes in the environment[12,13,37]. Furthermore, understanding structure-reactivity relationships for Ln³⁺ in nanoconfined systems can enable the separation of these critical elements using reactive nanopores[38]. The presented work illustrates that the energetics and products of adsorption could be predictably controlled by changing the size of a reactive nanopore.

## Results and discussion

**Adsorption complexes on corundum and nanoconfined Al₂O₃ surfaces.** The local coordination environment of adsorbed Nd, Tb, and Lu on corundum and nanoconfined Al₂O₃ surfaces was characterized using XAFS. We found that nanoconfinement promotes inner-sphere adsorption for Nd³⁺ cations and causes subtle elongation of the Lu–O bonds for inner-sphere Lu complexes. There is a stark difference in the surface speciation of adsorbed Nd when compared to Lu: Lu forms inner-sphere complexes (chemisorption) on both corundum and nano-confined Al₂O₃ surfaces. Nd, however, only forms outer-sphere complexes (physisorption) on corundum, and inner-sphere complexes on porous Al₂O₃ surfaces (Fig. 1). This conclusion is based on the Nd L_III-edge Fourier transformed XAFS spec-trum for corundum Al₂O₃ having no detectable 2nd shell neighbor, while for Nd adsorbed onto nanoconfined Al₂O₃ surfaces the 2nd shell due to Nd-Al backscattering is well-resolved in the spectrum (Fig. 1). This observation supports our hypothesis that a cation's $\Delta G_{hydr}$ defines the extent to which nanoconfinement affects its adsorption products. In the exam-ined set of cations, Nd³⁺ has the least exothermic $\Delta G_{hydr}$ (−3336 kJ·mol⁻¹)[39], and shows the most pronounced difference in the adsorption products when unconfined Al₂O₃ surfaces are compared to Al₂O₃ nanopores. On the other hand, Lu³⁺, which has the most exothermic $\Delta G_{hydr}$ (−3589 kJ·mol⁻¹)[39], produces nearly identical XAFS spectra when corundum and nano-confined Al₂O₃ surfaces are compared. All shell-by-shell fitting results for Nd, Tb, and Lu XAFS data are shown in Table 1. Raw data plotted in k-space with k-weight of 3 is shown in Fig. S1 in the Supporting Information file.

The 1st Nd–O shell for both corundum and nanoconfined Al₂O₃ surface was fit with a combination of two Nd–O theoretical backscattering paths at 2.45 ± 0.01 Å and 2.63 ± 0.01 Å (Table 1). The average Nd–O distance in these samples (2.54 Å) is similar to that reported for Nd³⁺ adsorbed onto SiO₂ surfaces nanocon-fined within 4 nm to 7 nm pores[29]. The 2nd shell due to Nd-Al backscattering (observed only for nanoconfined Al₂O₃ surface) was fit with a Nd-Al theoretical path at 3.46 ± 0.05 Å, consistent with a bi-dentate bi-nuclear complex geometry.

For Tb, XAFS data was collected only for nanoconfined Al₂O₃ surfaces, where Tb forms inner-sphere surface complexes. The 1st Tb–O shell was fit with a combination of two Tb–O theoretical paths at 2.33 ± 0.01 Å and 2.46 ± 0.01 Å, again showing a similar local environment to that of Tb adsorbed onto SiO₂ surfaces nanoconfined within 4 nm and 7 nm SiO₂ pores[29]. The 2nd shell fits with a Tb-Al theoretical backscattering path at 3.4 ± 0.01 Å, which indicates a bi-dentate bi-nuclear surface complex geome-try. XAFS data and fits for Tb, including fitting isolated 2nd shell, is shown in the Supporting Information file, Fig. S2. Similar to our earlier reported observations for Cu(II) and Ln(III) adsorbed on nanoconfined SiO₂ surfaces[29,30], dimerization reactions were evident for Tb and Lu on Al₂O₃: there is evidence of Tb-Tb and Lu-Lu backscattering contributions to the collected XAFS spectra. The presence of Tb–Tb backscattering suggests that some (not all) adsorbed Tb forms dimers or other types of polymeric species on nanoconfined Al₂O₃ surfaces; the Tb–Tb distance of

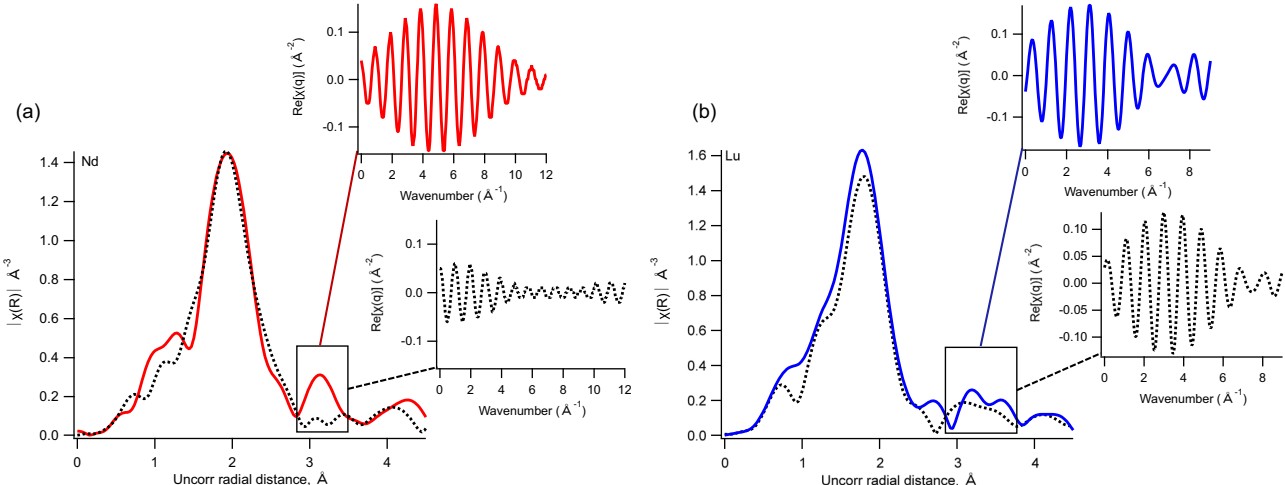

**Fig. 1 X-ray absorption fine structure spectroscopy data for neodymium and lutetium adsorbed onto non-porous and porous alumina.** X-ray absorption fine structure data for Nd (**a**) and Lu (**b**) adsorbed on corundum (dashed lines) and nanoconfined $Al_2O_3$ surfaces (solid lines); Fourier transform of each spectrum is shown. The insets in each panel illustrate back-transformed spectra of the isolated 2nd shell. For Nd, the 2nd shell is observed in the spectrum for the nanoconfined $Al_2O_3$ surface and corresponds to Nd-Al backscattering due to inner-sphere complexation; No 2nd shell is present for Nd adsorbed on corundum. For Lu, both spectra for corundum and nanoconfined $Al_2O_3$ are showing evidence for inner-sphere complexation.

**Table 1 Summary of X-ray absorption fine structure (XAFS) spectroscopy shell-by-shell fitting results for $Nd^{3+}$, $Tb^{3+}$, and $Lu^{3+}$ adsorbed onto corundum and nanoconfined $Al_2O_3$ with 4.4 nm pores.**

| Sample, coverage | [a]k-range | [a]R-range (Å) | Shell | [b]CN | [c]R (Å) | [d]$\sigma^2$ (Å²) | [e]$\Delta E_0$ eV | [f]R-factor | [g]Red $\chi^2$ | [h]Ind. Pts. |
|---|---|---|---|---|---|---|---|---|---|---|
| Nd-Al₂O₃- corundum, 13 mmol m⁻² | 2.6–10 | 1.5–3.8 | Nd-O | 5.4 ± 1.2 | 2.45 ± 0.01 | 0.001 ± 0.001 | 6.4 ± 1.2 | 0.011 | 4.3 | 11 |
| | | | Nd-O | 5.7 ± 1.0 | 2.63 ± 0.01 | 0.001 ± 0.001 | | | | |
| Nd-Al₂O₃-4 nm, 0.35 mmol m⁻² | 2.6–10 | 1.5–3.8 | Nd-O | 6.0 ± 0.6 | 2.46 ± 0.01 | 0.002 ± 0.001 | 5.9 ± 1.0 | 0.010 | 11 | 18 |
| | | | Nd-O | 5.2 ± 0.6 | 2.63 ± 0.01 | 0.003 ± 0.002 | | | | |
| | | | Nd-Al | 1.8 ± 0.7 | 3.47 ± 0.05 | 0.002 ± 0.004 | | | | |
| Tb-Al₂O₃-4 nm, 0.48 mmol m⁻² | 2.6–10 | 1.5–4.0 | Tb-O | 6.4 ± 1.0 | 2.36 ± 0.01 | 0.002 ± 0.002 | 6.3 ± 1.6 | 0.012 | 126 | 19 |
| | | | Tb-O | 4.6 ± 1.2 | 2.50 ± 0.02 | 0.003 ± 0.003 | | | | |
| | | | Tb-Al | 1.8 ± 1.0 | 3.38 ± 0.08 | 0.005 ± 0.018 | | | | |
| | | | Tb-Tb | 1.6 ± 2 | 3.64 ± 0.2 | 0.011 ± 0.030 | | | | |
| Lu-Al₂O₃- corundum, 38 mmol m⁻² | 2.6–10 | 1.4–4.0 | Lu-O | 5 ± 1 | 2.22 ± 0.02 | 0.005 ± 0.002 | 7.5 ± 0.6 | 0.013 | 28 | 19 |
| | | | Lu-O | 4.9 ± 0.7 | 2.34 ± 0.01 | 0.003 ± 0.002 | | | | |
| | | | Lu-Al | 4 ± 2 | 3.74 ± 0.06 | 0.006 ± 0.008 | | | | |
| | | | Lu-Lu | 5 ± 4 | 3.88 ± 0.06 | 0.007 ± 0.008 | | | | |
| Lu-Al₂O₃-4 nm, 0.59 mmol m⁻² | 2.6–10 | 1.4–4.0 | Lu-O | 5.9 ± 0.4 | 2.26 ± 0.01 | 0.003 ± 0.001 | 6.5 ± 0.6 | 0.007 | 65 | 19 |
| | | | Lu-O | 4.4 ± 0.6 | 2.40 ± 0.02 | 0.003 ± 0.002 | | | | |
| | | | Lu-Al | 3 ± 2 | 3.82 ± 0.06 | 0.009 ± 0.010 | | | | |
| | | | Lu-Lu | 4 ± 4 | 3.92 ± 0.08 | 0.011 ± 0.010 | | | | |

Fitting was done in R-space with simultaneous fitting of k-weights 1, 2, and 3. The amplitude reduction factor S₀ was set at 0.88 for Nd, 0.67 for Tb, and 0.71 for Lu, based on fitting XAFS spectra for model compounds Nd₂O₃, Tb₂O₃, and Lu₂O₃ with known structures. Errors at a 95% confidence level (2 sigma values) are shown.
[a]Usable k-range and R-range (uncorrected distances)
[b]Coordination number
[c]Bond length
[d]Debye-Waller factors: mean-square amplitude reduction factor, including thermal and static disorder components
[e]Energy shift between the theoretical and measured spectrum

[f]R-factor (mean square misfit) $R_{factor} = \frac{\sum_i (data_i - fit_i)^2}{\sum_i data_i^2}$

[g]Reduced chi-square $\chi_v^2 = \frac{N_{idp}}{N_{pts}} \sum_i \left(\frac{data_i - fit_i}{\varepsilon_i}\right)^2 / (N_{idp} - N_{var})$

[h]Independent points (number of data points minus the number of variable parameters) $N_{idp} = N_{pts} - N_{var}$

3.61 ± 0.06 Å suggests that the Tb polymerization is in the form of edge-sharing moieties (e.g., as in the structure of xenotime[40]).

Similar to Nd and Tb, two Lu–O backscattering paths were required to fit the 1st shell of the Lu spectra. For corundum, the Lu-O distances are 2.23 ± 0.02 Å and 2.34 ± 0.01 Å. For nanoconfined $Al_2O_3$ surfaces, the Lu-O distances are slightly longer, at 2.26 ± 0.01 Å and 2.41 ± 0.02 Å (Table 1). The elongation of the Lu-O distance under nanoconfinement may indicate that the local pH inside $Al_2O_3$ pores is lower than the controlled/measured pH

of the adsorption reactor (6.0 ± 0.1). In our earlier publication, we show that for Lu adsorbed onto $SiO_2$ surface at pH 4.0 the Lu-O distances are ~0.1 Å longer, compared to an analogous sample at pH 6.0, likely due to the lower abundance of $OH^-$ in the 1st shell around Lu at lower pH and shorter $Lu-OH^-$ distances compared to the $Lu-H_2O$ distance[29]. Recent studies indicate that inside $SiO_2$ nanopores protons are concentrated, driven by the negative surface charge inside nanopore[41]. However, $Al_2O_3$ surfaces are expected to be positively charged[42,43] at the near-neutral pH of

our experiments. Therefore, Lu-O elongation may be happening due to the EDL overlap and corresponding changes to the structure (hydrogen bonding) in nanoconfined water, rather than a higher proton concentration inside nanopores. The 2nd shell for Lu adsorbed onto corundum and nanoconfined $Al_2O_3$ surfaces was fit with Lu-Al theoretical backscattering path at 3.8 ± 0.06 Å, indicating a bi-dentate bi-nuclear adsorption complex, and additional Lu-Lu backscattering contribution indicating Lu dimers or other types of polymer surface species. The Lu–Lu distance of ~3.9 Å indicates a corner-sharing arrangement for Lu polyhedra (as we discuss in the previous publication[29], Lu-Lu edge-sharing would result in a shorter distance at ~3.55 Å, as in keiviite structure[44]). Similar to Lu adsorption on $SiO_2$, for $Al_2O_3$ we also observe bi-dentate complexation. On iron oxides, however, Lu tends to form mono-dentate complexes at pH 8 on hematite and at pH 5 on ferrihydrite[45].

Cation adsorption studies on nanoconfined $Al_2O_3$ surfaces are extremely limited and often lack spectroscopic analyses detailing surface speciation. On unconfined corundum surfaces uranium U(VI) likely adsorbs as outer-sphere complex because U(VI) adsorption was found to be both pH- and ionic-strength-dependent. However, on nanoconfined $Al_2O_3$ within ~1.3 nm pores U(VI) likely adsorbs as inner-sphere complexes since the adsorption is pH-dependent, while is independent of ionic strength[42]. These assumptions are also confirmed by sequential desorption experiments, where U(VI) shows irreversible adsorption onto nanoconfined $Al_2O_3$ and fully-reversible adsorption on corundum surfaces[42]. The most common U(VI) species at near-neutral pH is $UO_2CO_3$[46], for which $\Delta G_{hydr}$ was quantified at −41.17 kcal·mol$^{-1}$ (−172.3 kJ·mol$^{-1}$)[47]. The −172.3 kJ·mol$^{-1}$ $\Delta G_{hydr}$ value for U(VI) is by far less favorable than that for $Nd^{3+}$ (−3336 kJ·mol$^{-1}$); therefore macroscopically-observed differences in the U(VI) adsorption onto porous vs. non-porous $Al_2O_3$ agree with our predictions and observations for lanthanides.

**Heats of $Ln^{3+}$ adsorption quantified with *operando* flow microcalorimetry.** Heats of $Ln^{3+}$ adsorption on corundum and nanoconfined $Al_2O_3$ surfaces were quantified using *operando* flow microcalorimetry. At the conditions of these experiments, for all three $Ln^{3+}$, nanoconfinement reverses the enthalpic sign from an *endothermic* signal for the non-porous corundum to an *exothermic* signal on nanoconfined $Al_2O_3$ surfaces (Fig. 2). This finding is consistent with our previous investigations of $Cu^{2+}$ and $Ln^{3+}$ adsorption unto porous $SiO_2$ surfaces[29,30] whereby nanoconfinement resulted in a reversal of the enthalpic sign in the flow-through experiments. Additionally, *operando* flow microcalorimetry data can also point to the nature of surface complexes: e.g., the adsorption of $Cr^{3+}$ as inner-sphere complexes is exothermic for $SiO_2$ (quartz) and corundum, while the adsorption of $Al^{3+}$ on the same surfaces as outer-sphere complexes is endothermic[48].

In comparing the two $Al_2O_3$ surfaces, the adsorption of $Ln^{3+}$ was significantly more energetic on nanoconfined $Al_2O_3$ surfaces than on corundum surfaces, potentially indicating enhanced inner-sphere complexation. The summary of the microcalorimetric results ($Q_{ads}$ in mJ·m$^{-2}$ and $\Delta H_{ads}$ in kJ·mol$^{-1}$) and surface coverages (in mol·m$^{-2}$) is shown in Table 2. The molar enthalpies, $\Delta H_{ads}$, for $Lu^{3+}$, $Tb^{3+}$, and $Nd^{3+}$ were calculated to be –34.8, –13.4 and –55.3 kJ·mol$^{-1}$ on nanoconfined $Al_2O_3$ surfaces, and +1.77, +0.66, and +0.62 kJ.mol$^{-1}$ on non-porous corundum surfaces, respectively. The largest difference between both surfaces was observed for $Nd^{3+}$, which is consistent with our XAFS data and our hypothesis that the lightest lanthanides with the least exothermic $\Delta G_{hydr}$ are affected more by

nanoconfinement. It is crucial to note that no detectable calorimetric signal was detected at first for non-porous $Al_2O_3$, indicating that $Q_{ads}$ was ~0 mJ·m$^{-2}$. To achieve a detectable calorimetric signal, the microcalorimetry experiments on non-porous $Al_2O_3$ were conducted at a higher $Ln^{3+}$ concentration, resulting in higher surface coverages for non-porous corundum than for porous $Al_2O_3$. It is therefore possible that the overall higher energy measured for nanoconfined $Al_2O_3$ is disproportionately influenced by the high-energy sites, which are typically occupied first and can contribute more to the overall signal at lower surface coverages.

**Enthalpy, entropy, and Gibbs Free energy of adsorption measured in temperature-controlled batch experiments.** To assess the impact of surface coverage on the adsorption energetics we measured adsorption equilibrium constants in batch samples at temperatures ranging from 25 ± 1 °C to 45 ± 1 °C (298 K to 318 K). The surface coverages for all samples are provided in Table 2. In all batch reactors, the adsorbed Ln amount increases with increasing temperature, indicating endothermic adsorption for all $Ln^{3+}$ cations on porous and non-porous $Al_2O_3$. The dataset used for thermodynamic calculations is included in the Supporting Information file (Table S1). While the calculated $\Delta H_{ads}$ values were positive for both solids, $\Delta S_{ads}$ values were negative for $Al_2O_3$ nanopores, and positive for corundum surfaces (Table 2). This result indicates a higher degree of freedom for species adsorbed onto the corundum surface and increased ordering of surface complexes inside nanopores. Due to these entropic effects, the calculated $\Delta G_{ads}$ values at room temperature are negative for corundum, indicating a spontaneous reaction, and positive for $Al_2O_3$ nanopores, indicating that adsorption is non-spontaneous.

When all batch and flow samples are considered, $\Delta H_{ads}$ values show a dependency on the surface coverage up to ~0.5 mmol m$^{-2}$, at which point mono-layer coverage is achieved (Table 2, Fig. 3). With increasing surface coverages $\Delta H_{ads}$ values become more endothermic. This finding is not surprising, as we noted above, since at lower surface coverages the adsorption predominantly occurs at high-energy sites; and with increasing surface coverage, lower energy sites become progressively occupied. All data from the batch and *operando* microcalorimetry experiments is summarized in Table 2 and plotted in Fig. 3 to illustrate this point.

Direct calorimetric measurements of adsorption enthalpies as a function of the amount adsorbed remain limited. However, using titration calorimetry, Machesky et al.[49] found that for the adsorption of iodate, fluoride, phosphate, and salicylate onto goethite, adsorption enthalpies become less exothermic as surface coverage increases, while at high surface coverages (>70%) even become endothermic for phosphate and fluoride. More recently Falaise et al.[50] observed variations of adsorption enthalpies with surface coverage for the sorption of $Th^{4+}$, $[UO_2]^{2+}$, and $Nd^{3+}$ in a porous metal-organic framework, although each cation exhibited a different trend depending on its adsorption process.

**Interpreting data on structure and energetics together.** The overall free energy of adsorption is the sum of the electrostatic and chemical free energy changes (favorable to adsorption) and the solvation free energy change (unfavorable to adsorption)[51]:

$$\Delta G_{ads} = \Delta G_{coul} - \Delta G_{hydr} + \Delta G_{chem} \qquad (1)$$

Consider electrostatic interactions ($\Delta G_{coul}$) first: in our experiments, the $Al_2O_3$ surfaces are positively charged as they are in solutions at pH 6.0, which is below the reported point of zero charge (pH$_{PZC}$) values for various porous and non-porous

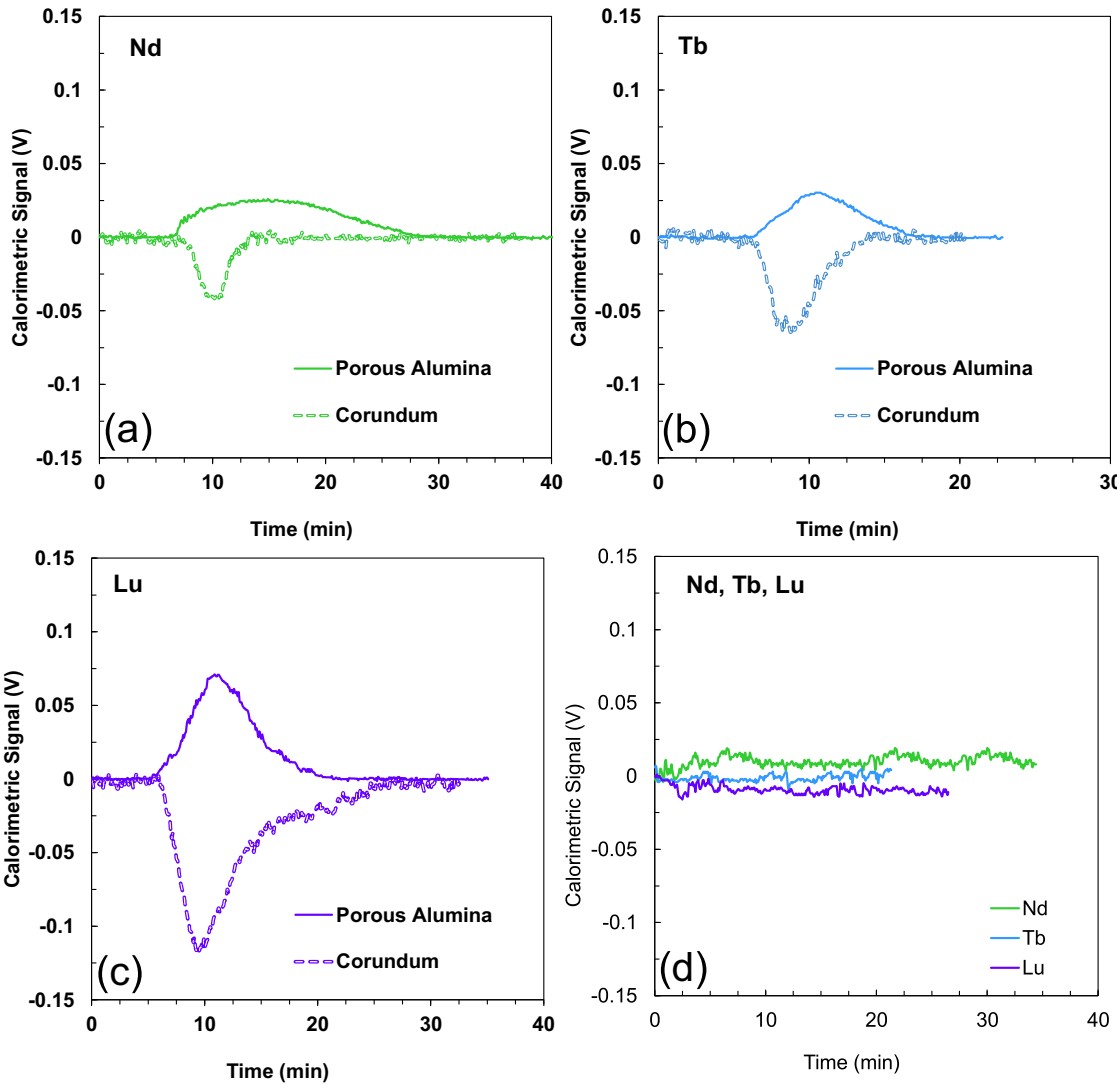

**Fig. 2 *Operando* microcalorimetry data for neodymium, terbium, and lutetium adsorbing onto non-porous and porous alumina.** Calorimetric signal obtained for the complexation of (**a**) $Nd^{3+}$, (**b**) $Tb^{3+}$ and (**c**) $Lu^{3+}$ on corundum and $Al_2O_3$ surfaces nanoconfined within 4 nm pores. An increase in voltage resulting in a positive peak corresponds to a release of energy and hence an exothermic reaction. For porous $Al_2O_3$, the concentrations were 7.86 μM for $Nd(NO_3)_3$, 11.22 μM for $Tb(NO_3)_3$ and 8.42 μM for $Lu(NO_3)_3$. Due to the low calorimetric signal obtained for corundum at the same aqueous concentrations (**d**), the concentrations of the stock solutions were increased to 157.2 μM for $Nd(NO_3)_3$, 224.4 μM for $Tb(NO_3)_3$, and 168.4 μM for $Lu(NO_3)_3$. This data is not normalized to the mass of solids used nor is it obtained for the same $Ln^{3+}$ concentrations. See text and Table 2 for normalized energies.

$Al_2O_3$ phases. The $pH_{PZC}$ values are 8.7 for $\gamma$-$Al_2O_3$ with 2 nm pores[43], 10.9 for porous $Al_2O_3$ with ~1.3 nm pores[42], and 9.7 for corundum[42]. Lanthanides are Brønsted acids and can hydrolyze water; however, $pK_a$ values for the first hydrolysis product for the $Ln^{3+}$ cations considered here are >7[52], thus in our experiments, $Ln^{3+}$ cations are expected to be in the aqua-ion form $[Ln·(H_2O)_8]^{3+}$. Therefore, the coulombic interactions at $Al_2O_3$ surfaces are unfavorable for adsorption and are likely more unfavorable for $Nd^{3+}$ because of its larger solvation shell and less effective charge screening compared to $Lu^{3+}$. Consider hydration energies ($\Delta G_{hydr}$) second: the $\Delta G_{hydr}$ for $Nd^{3+}$ is lower than that of $Lu^{3+}$, which is then harder to de-solvate prior to inner-sphere adsorption. Consider chemical free energy changes ($\Delta G_{chem}$) third: the Lu-O bond length is shorter by ~0.2 Å compared to Nd-O bond length (Table 1); therefore, $\Delta G_{chem}$ for Lu surface complexes at Al-OH sites is more favorable than for the analogous Nd complexes. We observed that $Lu^{3+}$ undergoes chemisorption on both confined and unconfined surfaces, while

$Nd^{3+}$ undergoes physisorption on unconfined, and chemisorption on nanoconfined surfaces. We conclude that the $\Delta G_{chem}$ term for $Lu^{3+}$ compensates for the more unfavorable $\Delta G_{hydr}$ contribution in both nanoconfined and unconfined systems. For $Nd^{3+}$, because of its lower degree of charge screening and less favorable $\Delta G_{chem}$, physisorption dominates for unconfined surfaces. We interpret the switch from outer- to inner-sphere adsorption for Nd in nanopores to be due to $\Delta G_{hydr}$ becoming less negative due to nanoconfinement, which allows $Nd^{3+}$ to shed 1–2 $H_2O$ molecules prior to adsorption in nanopores. It is important to note, that *all of the considered $\Delta G$ values are likely affected by nanoconfinement in different ways*. Solvation free energies become less negative in nanopores, compared to the reported $\Delta G_{hydr}$ values[29], however, the exact change in the $\Delta G_{hydr}$ value remains unresolved. Similarly, $\Delta G_{coul}$ likely increases in nanopores because of decreased charge screening due to $\Delta G_{hydr}$ values becoming less negative. Future work is urgently needed to unravel all the important thermodynamic contributions in charged

**Table 2 Summary of thermodynamic parameters calculated from batch adsorption data and measured by *operando* flow microcalorimetry experiments.**

| | Coverage, μmol·m⁻² | | $\Delta H_{ads}$, kJ·mol⁻¹ | | $\Delta S_{ads}$, kJ·mol⁻¹K | | $\Delta G_{ads}$ at 25 C, kJ·mol⁻¹ | |
|---|---|---|---|---|---|---|---|---|
| | CRDM | 4nm-Al₂O₃ | CRDM | 4nm-Al₂O₃ | CRDM | 4nm-Al₂O₃ | CRDM | 4nm-Al₂O₃ |
| Batch adsorption measurements | | | | | | | | |
| Nd | 0.002 | 70 | +6 | +18 | +0.02 | −0.02 | −1 | +25 |
| Tb | 0.003 | 211 | +1 | +9 | +0.01 | −0.04 | −2 | +22 |
| Lu | 0.002 | 505 | +4 | +18 | +0.02 | −0.01 | −3 | +20 |
| Lu | | 102 | | +3 | | −0.07 | | +23 |
| | Coverage, μmol·m⁻² | | $\Delta H$, kJ·mol⁻¹ | | | | | |
| | CRDM | 4nm-Al₂O₃ | CRDM | 4nm-Al₂O₃ | | | | |
| Operando flow microcalorimetry measurements | | | | | | | | |
| Nd | 25 | 0.00777 | +1 | −55 | | | | |
| Tb | 34 | 0.14578 | +1 | −13 | | | | |
| Lu | 38 | 0.01401 | +2 | −35 | | | | |

CRDM = corundum.

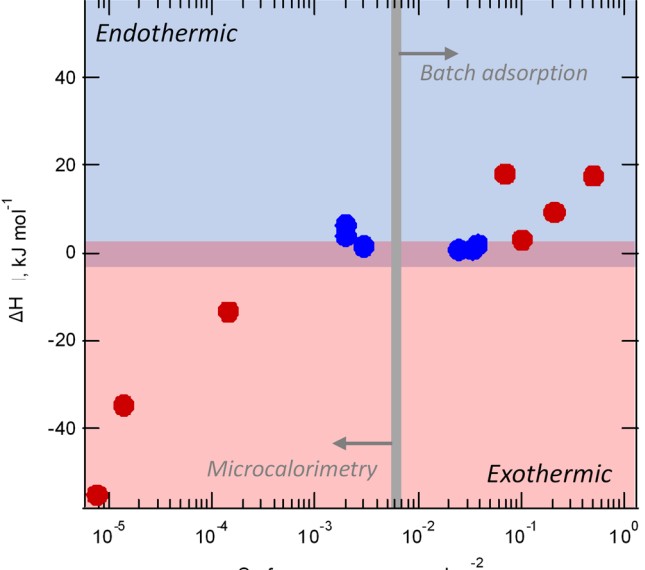

**Fig. 3 All measured enthalpies plotted as a function of surface coverage.** Red symbols are Al₂O₃ nanopores, blue symbols are corundum. Data at <0.001 mmol m⁻² coverage is from microcalorimetry measurements, and data for higher surface coverages is from batch adsorption measurements.

nanopores, that control surface reactivities in important ways presented here.

## Conclusions

In summary, we show that an ion's $\Delta G_{hydr}$ determines whether nanoconfinement changes the products of its adsorption and associated thermodynamics, and therefore $\Delta G_{hydr}$ can be used as a predictor. Since $Nd^{3+}$ has less negative $\Delta G_{hydr}$ compared to $Lu^{3+}$, it forms different surface complexes on unconfined versus nanoconfined Al₂O₃ surfaces. The balance between an ion's (de)solvation costs, coulombic interactions, and chemical free energy change dictates whether adsorption will occur through an outer- or an inner-sphere complex. Therefore, in nanoconfined systems where the average dielectric response of water is lowered[19], inner-sphere complexation is promoted. This is because a decreased dielectric response makes $Ln^{3+}$ $\Delta G_{hydr}$ less negative, reducing the energetic cost of the ion's partial desolvation prior to inner-sphere adsorption.

## Methods

**Al₂O₃ materials and temperature-controlled batch adsorption experiments.** Alumina Al₂O₃ with a mean pore diameter of $4.4 \pm 0.1$ nm and surface area of $335 \pm 2$ m² g⁻¹ (herein referred to as "nanoconfined Al₂O₃ surfaces") and non-porous corundum α-Al₂O₃ particles with a surface area of 1.5 m² g⁻¹ were purchased from Sigma Aldrich. Lanthanide stock solutions were made from nitrate salts Ln(NO₃)₃ using 18 MΩ·cm Milli-Q water.

To determine enthalpy ($\Delta H_{ads}$), entropy ($\Delta S_{ads}$), and Gibbs free energy ($\Delta G_{ads}$) of $Ln^{3+}$ adsorption samples were prepared by shaking $20 \pm 1$ mg of dry Al₂O₃ powders in ~19 mL of 0.01 M HEPES (N-(2-Hydroxyethyl)piperazine-N′-(2-ethanesulfonic acid)) buffer solution for 1 hour prior to adding $Ln^{3+}$. To begin the adsorption experiment, ~1 mL of $Ln^{3+}$ stock solution was added to each reactor to obtain a 20 mL total volume at a $Ln^{3+}$ concentration of 0.03-0.34 μM. The pH was $6.5 \pm 0.1$ for all samples. Adsorption proceeded for 23 hours at controlled temperatures of $25 \pm 1$ °C, $35 \pm 1$ °C, and $45 \pm 1$ °C in a water bath. Samples were withdrawn while still submerged in the water bath and filtered using a 0.2 μm nylon membrane immediately, so no temperature changes occur during sampling. Samples were acidified with 6 N ultrapure HNO₃. The concentrations of each $Ln^{3+}$ remaining in the solution after adsorption took place were quantified using inductively coupled plasma mass spectrometry (ICP-MS, NexION 350D, Perkin Elmer). Calibration curves for each analyte were obtained by running certified standard solutions prior to each analytical run, with a resulting R² value of 0.9999 or better. After equilibrium $Kd$ values were determined from aqueous concentrations, enthalpy, and entropy values were then calculated using van't Hoff Eq. (2); and free energy was calculated using Eq. (3):

$$\ln(K_d) = \Delta S/R - \Delta H/RT \qquad (2)$$

$$\Delta G = \Delta H - T\Delta S \qquad (3)$$

where $K_d$ is the equilibrium constant for a given temperature and $Ln^{3+}$ concentration, $T$ is the absolute temperature (K), and $R$ is the universal gas constant (8.314 J·mol⁻¹·K⁻¹). Plotting $\ln(K_d)$ vs $1/T$ yielded straight lines for both Al₂O₃ solids and for all $Ln^{3+}$, and we used the slope and intercept values to estimate $\Delta H_{ads}$, $\Delta S_{ads}$, and $\Delta G_{ads}$.

**X-ray absorption fine structure analysis.** Samples for the XAFS analyses were prepared by shaking $400 \pm 5$ mg of dry Al₂O₃ powders in 148 mL of 0.01 M NaCl for a minimum of 48 h. To begin the adsorption experiment, ~1 mL of $Ln^{3+}$ stock solution was added to each reactor to obtain a 150 mL total volume at a $Ln^{3+}$ concentration of 0.1 mM. These solutions were immediately adjusted to pH $6.0 \pm 0.1$ with NaOH or HCl. Throughout our experiments we chose pH 6.0 or pH 6.5 because this pH range is environmentally relevant: due to the atmospheric CO₂ dissolution into natural waters, pH between 6 and 6.5 is typical in natural systems. The second reason for choosing pH <7 is because it is below the first hydrolysis constant for lanthanides (>7), therefore the cations of interest were present as aqua-complexes with +3 charge in solution[52]. $Ln^{3+}$ adsorption proceeded for 48 h at ambient temperature (22 °C), at which point adsorption equilibrium was reached. Samples were then centrifuged, and the supernatant was filtered using a 0.2 μm nylon membrane filter before being acidified with 6 N ultrapure HNO₃. The concentrations of each $Ln^{3+}$ remaining in the solution after adsorption took place were quantified using ICP-MS, NexION 350D.

The remaining wet pastes from the bottom of the centrifuged reactors were stored in a refrigerator at 4 °C for XAFS spectroscopy analyses. Prior to data collection, these pastes were mounted on plastic sample holders with ~2 mm depth. XAFS data at the Nd, Tb, and Lu L_III-edges was collected using beamline 10-BM at the Advanced Photon Source (APS), Argonne National Laboratory. The beamline

is equipped with a water-cooled Si(111) monochromator, which was detuned by 50% to reject higher harmonics and calibrated using metal reference foils. The monochromator step size was 10 eV in the pre-edge, 0.5 eV in the XANES region, and 0.05 Å$^{-1}$ in the XAFS region. Fluorescent counts were collected using a Vortex Si Drift solid-state 4-element detector. The XAFS data were processed using the Athena interface and fitted using the Artemis interface[53] to the IFEFFIT[54] by fitting theoretical paths[55], which were based on the structures of lanthanide-containing apatite[56]. The structure files were edited to remove partial occupancies so that FEFF calculations could be performed. The background subtraction (AUTOBK algorithm[57]), normalization, and conversion into k-space were conducted as described elsewhere[58]. The fitting was done in R-space using a Hanning window and k-weights of 1, 2, and 3. R-factor cut-off of <0.05 was used to indicate a reasonable fit. In our samples, R-factors are between 0.01 and 0.02. The amplitude reduction factor ($S_0$) was determined by fitting XAFS spectra collected on $Nd_2O_3$, $Tb_2O_3$, and $Lu_2O_3$ standards; $S_0$ was 0.88 for Nd, 0.67 for Tb, and 0.71 for Lu. Fitting errors at a 95% confidence level (2 sigma) are shown in parenthesis in Table 1.

**Flow microcalorimetry experiments.** Custom flow microcalorimeter in the Kabengi Laboratory at Georgia State University[59] was used to obtain thermal signatures and subsequently, the heats of $Nd^{3+}$, $Tb^{3+}$, and $Lu^{3+}$ ion exchange using a uniformly-packed micro-column with $20.0 \pm 0.5$ mg porous $Al_2O_3$ sample or $50.0 \pm 0.5$ mg of corundum particles. The packed microcolumn was equilibrated with a 0.01 M $NaNO_3$ solution at a flow rate of $0.30 \pm 0.03$ mL min$^{-1}$ until a steady baseline indicative of thermal equilibrium was observed. A known mass of $Ln^{3+}$ was injected into the column with a controlled volume of $Ln(NO_3)_3$ stock solution. The calorimetric signal corresponding to the interaction of $Ln^{3+}$ with the $Al_2O_3$ surfaces was obtained following $Ln^{3+}$ injection. Once the thermal signal returned to the original baseline, the input solution was switched back to 0.01 M $NaNO_3$. For porous $Al_2O_3$, the concentrations were 7.86 μM for $Nd(NO_3)_3$, 11.22 μM for $Tb(NO_3)_3$ and 8.42 μM for $Lu(NO_3)_3$. Due to the low calorimetric signal obtained for corundum, the concentrations of the stock solutions were increased to 157.2 μM for $Nd(NO_3)_3$, 224.4 μM for $Tb(NO_3)_3$, and 168.4 μM for $Lu(NO_3)_3$. Aqueous concentrations of $Ln^{3+}$ in the column effluent samples were quantified using ICP-MS as described above. The mass of $Ln^{3+}$ retained at (and subsequently desorbed from) the surface was determined by a mass balance calculation between the mass of the injected $Ln^{3+}$ and the mass recovered in all effluents. The heats of reactions ($Q_{ads}$ in mJ·m$^{-2}$) and molar enthalpies ($\Delta H_{ads}$ in kJ·mol$^{-1}$) were calculated by integrating the calorimetric peaks that were converted to energy units (Joules) by calibration with calorimetric peaks of known energy inputs generated from a calibrating resistor placed inside the microcolumn. The solution pH was adjusted daily to pH $6.0 \pm 0.1$ using dropwise addition of 0.1 M $HNO_3$ and 0.1 M NaOH. Changes in total concentration and ionic strength resulting from pH adjustments were determined to be negligible.

## Data availability

The temperature-dependent adsorption data used to calculate thermodynamic values is included in the Supporting Information file (Table S1). The raw microcalorimetry and XAFS datasets generated during the current study are available from the corresponding author on reasonable request.

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

## Acknowledgements
The authors thank Y. Ding and A.W. Knight for help during XAFS data collection, P. Ilani-Kashkouli for collecting raw microcalorimetry data, and Tuan Ho and Kevin Leung for pre-submission review of this manuscript. This work was supported by the US Department of Energy, Office of Science, Office of Basic Energy Sciences, Chemical Sciences, Geosciences and Biosciences Division under Field Work Proposal # 23-015452.

X-ray Absorption Spectroscopy experiments were performed at Sector 10 at the Advanced Photon Source, an Office of Science User Facility operated for the U.S. Department of Energy (DOE) Office of Science by Argonne National Laboratory. This paper describes objective technical results and analysis. Any subjective views or opinions that might be expressed in the paper do not necessarily represent the views of the U.S. Department of Energy or the United States Government. This article has been authored by an employee of National Technology & Engineering Solutions of Sandia, LLC under Contract No. DE-NA0003525 with the U.S. Department of Energy (DOE). The employee owns all right, title and interest in and to the article and is solely responsible for its contents. The United States Government retains and the publisher, by accepting the article for publication, acknowledges that the United States Government retains a non-exclusive, paid-up, irrevocable, world-wide license to publish or reproduce the published form of this article or allow others to do so, for United States Government purposes. The DOE will provide public access to these results of federally sponsored research in accordance with the DOE Public Access Plan https://www.energy.gov/downloads/doe-public-access-plan. This paper describes objective technical results and analysis. Any subjective views or opinions that might be expressed in the paper do not necessarily represent the views of the U.S. Department of Energy or the United States Government.

## Author contributions
The manuscript was written through contributions of all authors. All authors have given approval to the final version of the manuscript. AGI developed research hypothesis, prepared samples, and performed XAFS experiments and data analysis. NK designed the microcalorimetry experiments, interpreted microcalorimetry and other presented results. JGS and KMMS assisted in sample preparation and initial calculations from the controlled-temperature batch adsorption experiments.

## Competing interests
The authors declare no competing interests.
