## [Peer Review File · Communications Chemistry]

Reviewers' comments:

Reviewer #1 (Remarks to the Author):

The manuscript deals with an interesting topic. The text is well written and fairly easy to read, but nevertheless there are a number of comments that should be eliminated before publication.

Experiment:

Al₂O₃ materials and temperature-controlled batch adsorption experiments

Why was the pH 6.5 chosen?

Since lanthanides are quite active in complexation, did the authors check the effect of the buffer on sorption? Has the sorption on the lanthanide filters been tested? It could have a strong effect on the sorption value (I mean K_d vs T)

Where can we see the data obtained to calculate the thermodynamic parameters

X-ray absorption fine structure analysis

Why pH 6 was chosen? The pH value should be written somewhere in the text, not only in the experimental part

The obtained EXAFS spectra should be present somewhere (may be in SI) in the k-space.

Adsorption complexes on corundum and nanoconfined Al₂O₃ surfaces

Table 1. Could the authors explain why they use so different parameters for the fitting[^] in particular K range and R range. For comparison the EXAFS spectra usually use the same condition for the whole dataset.

Why the ΔE_0 differs so much from sample to sample?

And where are the Tb spectra?

What about the influence of strong and weak sorption sites? How this fits in with the authors' hypotheses?

It would also be interesting to see the switching of sorption mechanisms for Nd not only on EXAFS data but also on experimental sorption data? Do the authors not have such data? That would be an excellent confirmation

Reviewer #2 (Remarks to the Author):

In this paper, the authors use X-ray absorption fine structure spectroscopy and operando flow microcalorimetry to determine nanoconfinement effects on the energetics and local coordination environment of trivalent lanthanides adsorbed on Al₂O₃ surfaces. The presented work illustrates that the energetics and products of adsorption could be predictably controlled by changing the size of a reactive nanopore. So, I suggest that this manuscript can be considered after minor revision.

1. The XAFS results on Nd-O, Tb-O, Nd-Al, Tb-Al and so on are reported in this work. What are the reference and criterion when these XAFS data are analyzed? They should be mentioned.
2. Some format problems should be careful. "148 ml, 150 ml" should be "148 mL, 150 mL".
3. The blank should be added before and after some symbols, such as "6.0±0.1".

Reviewer #3 (Remarks to the Author):

The manuscript by Ilgen et al describes experimental studies of lanthanide adsorption to free and confined mineral surfaces. Using XAFS and calorimetry techniques, the adsorption of Lu, Nd, Tb to nanoporous alumina and corundum were investigated. Important and very interesting differences between nanopore adsorption and adsorption to the free mineral surface are observed. The adsorption corundum is endothermic for all cations, while adsorption in nanopores was always exothermic. Subtle changes in the structure of surface complexes are also observed, which seem to indicate tighter surface bonding in nanopores. The properties of ions in nanopores are poorly understood and detailed, atomic level research is needed to understand these complex systems better. The experiments and analysis are carefully carried out and the conclusions are sound and novel.

Point-by-point response to the reviewers' comments,

Manuscript submitted to *Communications Chemistry*, title: "Ion solvation as a predictor of lanthanide adsorption structures and energetics in alumina nanopores", Anastasia G. Ilgen, Nadine Kabengi, Kadie Sanchez, and Jacob Smith

Reviewers' comments are shown in regular black font, and authors' responses are in *blue italics*.

May 28, 2023

Reviewer #1 (Remarks to the Author):

The manuscript deals with an interesting topic. The text is well written and fairly easy to read, but nevertheless there are a number of comments that should be eliminated before publication.

Response: We are grateful to the reviewer for their constructive feedback. To address the comments below, we re-fitted the data using the same k-value in the beginning of the range for all samples. The k-value for the end of the range was chosen based on the noise level in the data. The R-range was selected to include 1st and 2nd shells, therefore R-range varies slightly between the three elements since their Ln-O and Ln-Al distances vary slightly. For the same element we use the same R-range in the dataset. We hope that these revisions make our work acceptable for publication in Communications Chemistry.

Experiment:

Al₂O₃ materials and temperature-controlled batch adsorption experiments
Why was the pH 6.5 chosen?

Response: We chose pH 6.5 because it is environmentally relevant: due to the atmospheric CO₂ dissolution into natural waters, this pH is typical in natural systems. Second important reason for choosing pH <7 is because it is below the pKa values for lanthanides (>7), therefore they exist as aqua-complexes with a +3 charge in solution. We made a note about these two reasons in the manuscript text.

Since lanthanides are quite active in complexation, did the authors check the effect of the buffer on sorption?

Response: In the reported results, we only used HEPES buffer in the temperature-controlled experiments to avoid opening the vials and causing temperature fluctuations during the experiments. For microcalorimetry and XAFS samples the pH was adjusted with inorganic compounds as noted in the Methods section. Based on the reported studies for lanthanides, the complexation constant with HEPES buffer is measurable, but it is not significant, and the vast majority of a lanthanide in solution is predicted to exist as aqua-ions. For example, Eu(III) complexation constant with HEPES is $10^{-4.1}$ (Mandal et al, 2022).

Has the sorption on the lanthanide filters been tested? It could have a strong effect on the sorption value (I mean K_d vs T)

Response: We did not measure lanthanide adsorption onto nylon membrane filters, as such filtration is considered a "gold standard" for natural and synthetic samples intended for ICP-MS analysis. Based on

the previous studies, filtration through nylon filter with 0.2 μm pores removes 0.2% of lanthanide (lanthanum 3+) during sample filtration (Weltje et al., 2003). Our overall errors (including pipetting, weighting, dilution and ICP-MS analytical errors) are $\sim 3\%$, and the 0.2% error is included in this propagated error value since stock solutions containing lanthanides are also filtered in the same manner as the samples.

Where can we see the data obtained to calculate the thermodynamic parameters

Response: We included the dataset in the SI file.

X-ray absorption fine structure analysis

Why pH 6 was chosen? The pH value should be written somewhere in the text, not only in the experimental part

Response: We chose pH 6 because it is environmentally relevant: due to the atmospheric CO_2 dissolution into natural waters, this pH is typical in natural systems. Second important reason for choosing pH < 7 is because it is below the pK_a values for lanthanides (> 7), therefore they exist as aqua-complexes with +3 charge in solution. We made a note about these two reasons in the manuscript text.

The obtained EXAFS spectra should be present somewhere (may be in SI) in the k-space.

Response: We included all XAFS spectra plotted in k-space in the SI file.

Adsorption complexes on corundum and nanoconfined Al_2O_3 surfaces

Table 1. Could the authors explained why they use so different parameters for the fitting[^] in particular K range and R range. For comparison the EXAFS spectra usually use the same condition for the whole dataset.

Response: The noted differences in the selected ranges do not impact our fitting results. To address this comment, we adjusted our XAFS data fits to use the same k-range (2.6-10) for all samples. The R-range was selected to include the 1st and 2nd shells, therefore R-range varies slightly between the three elements since their Ln-O and Ln-Al distances vary slightly. For "sample sets" – e.g. the two Nd samples and the two Lu samples we use the same R-ranges for consistency.

Why the ΔE_0 differs so much from sample to sample?

Response: To address this comment, we re-visited Tb XAFS data analysis and adjusted ΔE_0 value to be more consistent with the other two lanthanides. This resulted in slightly longer bond lengths for all shells, which were increased by 0.03-0.04 \AA . In our final XAFS data fits the ΔE_0 values are similar across the whole series: 6.4 eV and 5.9 eV for Nd; 6.3 eV for Tb; and 7.5 eV and 6.5 eV for Lu.

And where are the Tb spectra?

Response: We added Tb spectra and corresponding fits to the SI file.

What about the influence of strong and weak sorption site? How this fits in with the authors' hypotheses?

The potential presence of sites with different affinities and therefore energetics does not alter the conclusions of the manuscript. Specifically, in flow microcalorimetry experiments, a return of the calorimetric signal to the initial baseline indicates a completion of the reaction, which is then assumed to have reached equilibrium. Therefore, the ΔH measured represents a summation of adsorption on the totality of the sites occupied at that particular concentration. The effect of confinement stands, particularly as it is backed up by changes in the surface complexes.

However, we cannot rule out the possibility that should the experiments on confined surfaces be run at higher concentrations (similar to those for corundum) a higher surface coverage and in fact a different equilibrium state may be obtained potentially sampling more sites, both with higher and lower energetics, and resulting in different magnitude of the ΔH .

It would also be interesting to see the switching of sorption mechanisms for Nd not only on exafs data but also on experimental sorption data? Do the authors not have such data? That would be an excellent confirmation

Response: We agree with the reviewer that an extensive investigation of lanthanide adsorption onto Al_2O_3 nanopores would be very interesting. We envision studies on pH- and ionic strength-dependent adsorption, as well as coverage-dependent adsorption thermodynamics to further clarify the phenomenon reported in our current manuscript; however, our research goal was to connect adsorption thermodynamics with molecular structures observed on the surfaces. Therefore, the extensive pH- and ionic-strength-dependency studies are outside of the current scope of work and constitute future research.

Reviewer #2 (Remarks to the Author):

In this paper, the authors use X-ray absorption fine structure spectroscopy and operando flow microcalorimetry to determine nanoconfinement effects on the energetics and local coordination environment of trivalent lanthanides adsorbed on Al_2O_3 surfaces. The presented work illustrates that the energetics and products of adsorption could be predictably controlled by changing the size of a reactive nanopore. So, I suggest that this manuscript can be considered after minor revision.

Response: Thank you for the careful review of our work and providing useful comment to further improve our manuscript.

1. The XAFS results on Nd-O, Tb-O, Nd-Al, Tb-Al and so on are reported in this work. What are the reference and criterion when these XAFS data are analyzed? They should be mentioned.

Response: To address this comment we added this sentence to our XAFS data analysis methodology: "R-factor cut-off of <0.05 was used to indicate a reasonable fit. In our samples R-factors are between 0.01 and 0.02." Additionally, our original and R1 submissions include the information about the analyzed standards and model compounds: "The XAFS data were processed using the Athena interface and fitted using the Artemis interface to the IFEFFIT by fitting theoretical paths, which were based on the structures of lanthanide-containing apatite. The structure files were edited to remove partial occupancies, so that FEFF calculations could be performed. The amplitude reduction factor (S_0) was determined by fitting XAFS spectra collected on Nd_2O_3 , Tb_2O_3 , and Lu_2O_3 standards; S_0 was 0.88 for Nd, 0.67 for Tb, and 0.71 for Lu."

2. Some format problems should be careful. "148 ml, 150 ml" should be "148 mL, 150 mL".

Response: We corrected all instances of “ml” to “mL”

3. The blank should be added before and after some symbols, such as “6.0±0.1”.

Response: We added a space before and after all “±” in all instances in text to improve readability.

Reviewer #3 (Remarks to the Author):

The manuscript by Ilgen et al describes experimental studies of lanthanide adsorption to free and confined mineral surfaces. Using XAFS and calorimetry techniques, the adsorption of Lu, Nd, Tb to nanoporous alumina and corundum were investigated. Important and very interesting differences between nanopore adsorption and adsorption to the free mineral surface are observed. The adsorption corundum is endothermic for all cations, while adsorption in nanopores was always exothermic. Subtle changes in the structure of surface complexes are also observed, which seem to indicate tighter surface bonding in nanopores. The properties of ions in nanopores are poorly understood and detailed, atomic level research is needed to understand these complex systems better. The experiments and analysis are carefully carried out and the conclusions are sound and novel.

Response: Thank you for reading our work and for your positive review.

References

Mandal P, Kretzschmar J, Drobot B, Not just a background: pH buffers do interact with lanthanide ions—a Europium(III) case study. *Journal of Biological Inorganic Chemistry* (2022) 27:249–260

Weltje L, Hollander WD, Wolterbeek Th, Adsorption of metals to membrane filters in view of their speciation in nutrient solution. *Environmental Toxicology and Chemistry*, Vol. 22, No. 2, pp. 265–271, 2003

Reviewer #1 (Remarks to the Author):

The new version of the manuscript may be accepted for publication

Reviewer #2 (Remarks to the Author):

The authors have solved most of the issues. I think this paper will be accepted.